# Research on Micro-Displacement Measurement Accuracy Enhancement Method Based on Ensemble NV Color Center

**DOI:** 10.3390/mi14050938

**Published:** 2023-04-26

**Authors:** Yuqi Liu, Zhonghao Li, Hao Zhang, Hao Guo, Ziyang Shi, Zongmin Ma

**Affiliations:** 1Key Laboratory of Instrument Science and Dynamic Testing Ministry of Education, North University of China, Taiyuan 030051, China; liuyuqi1831@163.com (Y.L.); zhanghao_19961004@163.com (H.Z.); guohaonuc@163.com (H.G.); shiziyangnuc@163.com (Z.S.); mzmncit@163.com (Z.M.); 2Key Lab of Quantum Sensing and Precision Measurement, Taiyuan 030051, China; 3Institute of Instrument and Electronics, North University of China, Taiyuan 030051, China

**Keywords:** quantum sensing technology, nitrogen-vacancy color center, micro-displacement, magnetic flux concentrator, permanent magnet, optically detected magnetic resonance

## Abstract

This paper builds a corresponding micro-displacement test system based on an ensemble nitrogen-vacancy (NV) color center magnetometer by combining the correlation between a magnetic flux concentrator, a permanent magnet, and micro-displacement. By comparing the measurement results obtained with and without the magnetic flux concentrator, it can be seen that the resolution of the system under the magnetic flux concentrator can reach 25 nm, which is 24 times higher than without the magnetic flux concentrator. The effectiveness of the method is proven. The above results provide a practical reference for high-precision micro-displacement detection based on the diamond ensemble.

## 1. Introduction

Micro-displacement detection technology is the key to testing the high degree of precision, stability, and reliability of precision measurement instruments [1,2,3]. At present, the existing micro-displacement detection technologies to achieve micro-displacement measurements include optical interference technology, image methods, piezoelectric ceramics, and micro-mechanical structure designs. Integrated optical interferometers for the micro-displacement measurement of ultra-small self-focusing fiber probes have a linear measurement range of 10 µm, a linearity of 1.36%, and a sensitivity to displacement of 8.8 mV/µm [4]. The capacitance grating sensor is used to measure the displacement, achieving measurement errors of less than 8 µm [5]. The mathematical relationship between in-plane displacement and out-of-plane displacement in a deformed image taken by the optical microscope is utilized for image spheroidization and differential theory. Through simulation and application experiments, it has been verified that the absolute error measured by this method under an optical microscope of 50 times magnification is less than 0.2 µm [6]. Piezoelectric ceramics use probe pressure feedback to achieve micro-displacement testing [7]. However, piezoelectric ceramics have a significant hysteresis effect, creep effect, and nonlinearity in traditional displacement detection. In displacement tests using micro-mechanical structures [8], the technology required to process micro-mechanical structures is complex, and the accuracy requirement is high, which leads to high experimental costs. It can be seen from the above studies that existing micro-displacement measurement systems need to improve their measurement accuracy and practicability further to meet the needs of micro-displacement measurements.

Nitrogen-vacancy (NV) color centers in diamonds have attracted widespread attention, as they have good optical stability at room temperature [9,10,11,12,13] and have been widely used in quantum control, magnetic imaging, and microwave detection [14,15,16,17]. Continuous-wave optical detection magnetic resonance (CW-ODMR) was used to measure a magnetic field gradient with high sensitivity, and micro-displacement detection based on changes in the magnetic field gradient could be realized [18,19]. Although directly increasing the magnetic field gradient can achieve more precise micro-displacement detection, introducing a strong magnetic field will limit the application range of this kind of displacement measurement. Therefore, it is necessary to develop a high-precision micro-displacement detection technology based on weak magnetic spatial variations to ensure the application range of magnetic detection.

In experiments, a magnetic flux concentrator (MFC) is usually used to amplify the magnetic field’s intensity to obtain higher magnetic sensitivity. Fescenko et al. designed an MFC with a diameter of 10 mm and a height of 10 mm on the upper surface of a round table and a diameter of 370 µm on the lower surface of a round table. The distance between the two MFCs was 43 µm, and the measured magnetic field amplitude was 250 times that of the original [20]. In addition, Leroy et al. used two cylinders with a length of 1 mm and a diameter of 20 mm as MFCs and obtained improved magnetic sensitivity through experimental tests. The magnification was about 1000 times [21]. Wang Lei et al. used the high-precision sensitivity mechanism of the electron spin effect on a magnetic gradient field combined with the relationship between the magnetic gradient field and micro-displacements to study the micro-displacement measurement method based on the electron-spin-sensitive magnetic mechanism of the diamond nitrogen-vacancy color center [22].

Therefore, drawing from the above experimental analyses, this experiment is based on the MFC and the electron-spin-sensitive magnetic mechanism of diamond nitrogen-vacancy color centers and aims to conduct micro-displacement measurement research. The effectiveness of the MFC in micro-displacement measurements is verified by comparing the results in the presence and absence of the MFC structure.

## 2. Materials and Methods

NV color centers in diamonds are obtained by replacing a carbon atom in the diamond lattice with a nitrogen atom. The structure composed of the nitrogen atom and adjacent carbon vacancies has C_3υ_ symmetry [23], and its principle structure is shown in Figure 1a. NV centers have two states, NV^0^ and NV^−^ [24]. The NV^−^ state has electronic polarization and optical readout abilities [25], which can be used for high-precision magnetic field detection. For the convenience of expression, the NV color centers referred to in this paper are NV^−^. The ground state of the energy level of the diamond NV color center is the triplet state of electron spin, as shown in Figure 1b. Under a zero magnetic field, the ground state of the NV m_s_ = ±1 is in a degraded state. There is a zero-field splitting of 2.87 GHz between the ground state of the NV and the energy level of m_s_ = 0 [26]. When there is an external magnetic field, m_s_ = +1 and m_s_ = −1 will degenerate, the electron spin energy level will undergo Zeeman splitting [27], and the corresponding resonance peak will split [28,29].

When the direction of the NV axis is *Z*′, the Hamiltonian of ground state electron spin of the NV color center is expressed as [30,31]:(1)$$H=h\left[DS_{Z^{\prime }}^{2}+E\left(S_{x^{\prime }}^{2}-S_{y^{\prime }}^{2}\right)+g\mu_{B}B\cdot S\right]$$
where *h* is the Planck constant, and the three items in square brackets are the zero-field splitting in the direction of the NV axis, the transverse zero-field splitting, and the Zeeman term under the action of the static magnetic field, respectively. *D* = 2.87 GHz is the zero-field splitting constant at room temperature, and *E* is the off-axis zero-field splitting constant. Without an external electric field, *E* = 0, and *Sx*′, *Sy*′, and *Sz*′ represent the angular momentum of the three axis directions of the spin of NV electrons, respectively. The spin coefficient is 1. $\mu_{B}$ is a Bohr magneton ($\mu_{B}\;$ = 1.4 MHz/Gauss), and *g* is the Landég factor (*g* = 2).

According to the third term in Equation (1), under the action of a static magnetic field, the corresponding resonant frequency *ω* can be expressed by [32]:(2)$$\omega =D\pm \frac{g\mu_{B}B}{h}$$

Therefore, the relationship between the change in the resonant frequency Δ*ω* and the change in the static magnetic field intensity Δ*B* can be expressed as:(3)$$\Delta \omega =\frac{g\mu_{B}\Delta B}{h}$$

In the experiment, the magnetic field was provided by a permanent magnet. The coordinate system was established with the cylindrical permanent magnet material as the surface center of the N_35_-sintered magnet and the coordinate origin axis direction as the Z-axis direction, as shown in Figure 1a. According to the symmetry of the cylinder, it is known that the magnetic field intensity in other directions will offset each other, which is only affected by the magnetic field’s intensity in the Z-axis direction. Therefore, the magnetic field intensity *B_Z_*_+_ at any point on the positive half-axis of the Z-axis is [33]:(4)$$B_{Z+}=\frac{M}{4\pi }\int \int_{S+}\frac{\left(z-Z\right)}{{\sqrt{{\left(\rho -x\right)}^{2}+y^{2}+{\left(z-Z\right)}^{2}}}^{3}}dS$$
where *M* is the constant magnetization of the permanent magnet, and *S*_+_ is the surface surrounding the positive half-axis of the Z-axis of the permanent magnet.

After adding the MFC, the changing relationship of magnetic field intensity can be expressed as:(5)$$B_{\delta Z+}=N\frac{M}{4\pi }\int \int_{S+}\frac{\left(z-Z\right)}{{\sqrt{{\left(\rho -x\right)}^{2}+y^{2}+{\left(z-Z\right)}^{2}}}^{3}}dS$$
where *N* is the MFC magnification, so the corresponding resonant frequency Δ*ω_δ_* can be expressed as [34]:

As shown in Figure 2, the magnetic field intensity of the permanent magnet is 43–45 mm along the Z-axis. There is almost a linear change in the distance. Assuming the linear coefficient *K*, the frequency difference is as follows:(6)$$\Delta \omega_{\delta }=\omega_{\delta 2}-\omega_{\delta 1}=\frac{g\mu_{B}\Delta B_{\delta Z+}}{h}=N\frac{g\mu_{B}K\Delta x}{h}$$
(7)$$\Delta \omega =\omega_{2}-\omega_{1}=\frac{g\mu_{B}\Delta B}{h}=\frac{g\mu_{B}K\Delta x}{h}$$

Assuming that the displacement variation with and without MFC is Δ*x*, the frequency difference ratio is:(8)$$\frac{\Delta \omega }{\Delta \omega_{\delta }}=\frac{\frac{g\mu_{B}\Delta B}{h}}{\frac{g\mu_{B}\Delta B_{\delta Z+}}{h}}=\frac{\frac{g\mu_{B}K\Delta x}{h}}{N\frac{g\mu_{B}K\Delta x}{h}}=\frac{1}{N}$$

Combining Equation (2) with Equation (8), it can be seen that with increasing flux concentration, the magnetic field gradient corresponding to the spin surface increases by *N* times and the corresponding resonant frequency increases by *N* times; therefore, high-precision displacement detection can be carried out under a weak magnetic field. The size radius of the permanent magnet used in this experiment is *r* = 5 mm, and the thickness is *h* = 2 mm.

The MFC was made from a permalloy material, and the MFC combined conical and cylindrical shapes [35]. The diameter of the cone bottom was 15 mm, and the height was 15 mm. The diameter of the cylinder was 2 mm; the height was 15 mm. The distance between the two parts was 2 mm, and the magnification of the simulated magnetic field was about 30 times.

In this experiment, the permanent magnet was fixed on the horizontal moving displacement table (the minimum step accuracy was 0.01 mm). Then, the central axis of the MFC was aligned with the axis of the cylindrical magnet; when the distance between the permanent magnet and the diamond sample was measured in the range of 43–45 mm, the magnetic field gradient changed linearly with or without the magnetic integrator. When measuring the magnetic field strength, the Model931 Gauss meter, produced by Qingdao Zhongyu Huantai Magnetoelectric Technology Co., Ltd., Qingdao, China was used, and the small displacement required by the horizontal step of the displacement table was read out in the Gauss meter, as shown in Figure 3a. The linear fitting shows that the magnetic field gradient is linear with changes in small displacement; additionally, the linearity is about 0.99, and the slope is dB_1_/dZ = −9.89 ± 0.22 Gauss/mm. At the same time, the magnetic field intensity of cylindrical magnets without MFC at the same distance was tested. The experimental results show that the linearity is about 0.99, and the slope is dB_2_/dZ = −0.402 ± 0.009 Gauss/mm, as shown in Figure 3b. It can be obtained from the experimental data that the change rate of magnetic field intensity with MFC is about 24 times that without the MFC, indicating that the MFC can significantly enhance changes in the magnetic field gradient on the surface of the NV color center.

The schematic diagram of the experimental test system is shown in Figure 4. In the experiment, the resonant microwave signal was emitted by the microwave source (N5183B MXGX series microwave analog signal generator, Keysight is headquartered in Santa Rosa, CA, USA) to promote the spin reversal of the diamond at the resonant frequency point, resulting in the weakening of the intensity of the fluorescence. The modulated signal generated by the microwave source was used for spin resonance to generate a modulated fluorescence signal. A 532 nm laser (Cnilaser, MLL-S-532 nm, Changchun New Industry Company, Changchun, China) emitted a laser signal, and the green laser was irradiated on the diamond surface through the mirror, lens group, and dichroic mirrors. The fluorescence signal generated by the NV color center ensemble was transmitted to the photodetector (APD130A2/M, Thorlabs Company, Newton, NJ, USA) through dichroic mirrors and the filter, and the collected fluorescence signal was input into the lock-in amplifier for demodulation. The demodulated fluorescence signal was connected to the oscilloscope to complete the observation of the corresponding first-order differential signal. The NV color center ensemble sample concentration was 3 ppm, and size was 2 mm × 2 mm × 0.5 mm. The permanent magnet that produced gradient changes was fixed on the horizontally and vertically adjustable precision displacement table. As shown in Figure 4, the magnetic concentrator was placed on both sides of the diamond symmetry. In the measurement process, the magnetic field step was changed by adjusting the horizontal displacement to realize the magnetic field gradient induced by the spin system.

## 3. Results and Discussion

During the experiment, different magnetic field gradient values were obtained by controlling the micro-displacement of the permanent magnet relative to the diamond and detecting magnetic gradient changes by measuring changes in the resonant frequency of the NV color center under different magnetic field intensities. During the experiment, the step displacement of the single test was 0.25 mm. The CW-ODMR experimental data under different displacements were obtained in the linear magnetic field region and the MFC, as shown in Figure 5a. The CW-ODMR without the MFC under the same micro-displacement condition is shown in Figure 5b. To ensure the consistency of the experimental comparison, the resonance point on the right side of the experimental data in Figure 5a,b was selected to obtain the correlation between Zeeman splitting and micro-displacement. The relationship between frequency difference and displacement is calculated by resonance frequency, as shown in Figure 5c. The linear correlation coefficient of the MFC is 0.99, dΔ(*ω*_1_)/d*Z* = −23.4 MHz/mm, and the linear correlation coefficient without the MFC is0.97, dΔ(*ω*_2_)/d*Z* = −0.91 MHz/mm, as shown in Figure 5d. According to Figure 3a,b and Figure 5c,d, the Zeeman splitting frequency is reduced with decreasing magnetic induction intensity.

In the experiment, the frequency of the modulation signal was set to 1 kHz, and the modulation amplitude was 1 V. Under the MFC, the phase-locked amplifier was used to scan the microwave frequency, and the CW-ODMR curve with the resonance frequency of 3.193 GHz was modulated and demodulated. The obtained modulation curve is shown in Figure 6a. It can be seen that the slope of the demodulation curve α_1_ is 0.55 V/MHz. According to the value of slope α_1_, the change in the Zeeman splitting frequency with the position of the permanent magnet can be expressed by the change in the fluorescence amplitude, namely, *U*_1_ = *α*_1_Δ*ω*_1_. The microwave frequency was fixed at 3.193 GHz, the permanent magnet moved within a range of ±0.25 mm, and the single-step displacement was 0.05 mm. By recording the voltage amplitude of the output signal of the phase-locked amplifier at different positions, the relationship curve between the voltage amplitude and the micro-displacement is obtained. The test results are shown in Figure 6b. In the absence of the MFC, the corresponding CW-ODMR modulation and demodulation curves are shown in Figure 6c. At this time, due to the slight gradient of the direct magnetic field, it can be seen that the demodulation curve generates two closely arranged demodulation curves. The right curve is evaluated, and it can be seen the slope of the corresponding demodulation curve *α*_2_ is 0.56 V/MHz, that is, *U*_2_ = *α*_2_Δ*ω*_2_. The microwave frequency was fixed at 2.8896 GHz, and the permanent magnet moved within a range of ±0.25 mm; the single-step displacement was 0.05 mm. The relationship between voltage amplitudes and micro-displacement curves is obtained by recording the voltage amplitudes of lock-in amplifier output signals at different positions. The test results are shown in Figure 6d.

In this position, the fluorescence jitter with and without the MFC is compared, and it can be seen that microwaves were output at a fixed frequency with and without the MFC. The frequency points were set to 3.193 GHz and 2.8896 GHz to obtain the noise signals, as shown in Figure 7a,b. By comparing the signal in Figure 7a with the MFC and the signal in Figure 7b without the MFC, it can be seen that the MFC system has no obvious influence on signal noise. Therefore, the deviation in system noise is 0.334 mV after calculation. According to the ratio of system noise to the sensitivity of the system with and without the MFC, the resolution of the displacement test system with the MFC is 25 nm.

## 4. Conclusions

Based on the diamond NV color center ensemble, this paper uses CW-ODMR technology, modulation, demodulation technology, and magnetic accumulation technology through the electron-spin magnetic-field-sensitive mechanism, which, combined with the magnetic gradient change process, allowed the micro-displacement precision measurement method to be verified. By comparing the test results with and without the MFC, it could be seen that the displacement detection resolution could be increased by 24 times with the MFC. It has been proved that the MFC effectively improves the accuracy of micro-displacement measurements. The above results provide a helpful reference for micro-displacement detection based on the NV color center ensemble.

## Figures and Tables

**Figure 1 micromachines-14-00938-f001:**
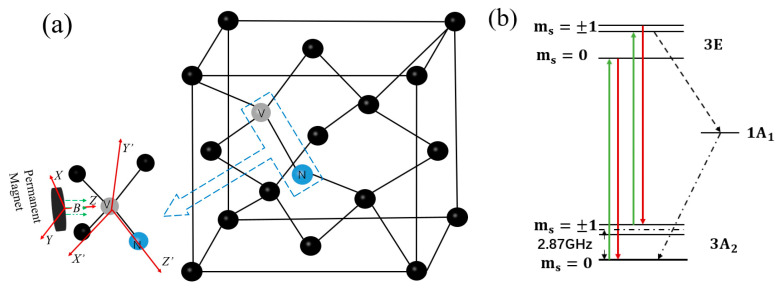
(**a**) Principle structure of diamond NV color center; (**b**) Electron spin triplet state of diamond NV.

**Figure 2 micromachines-14-00938-f002:**
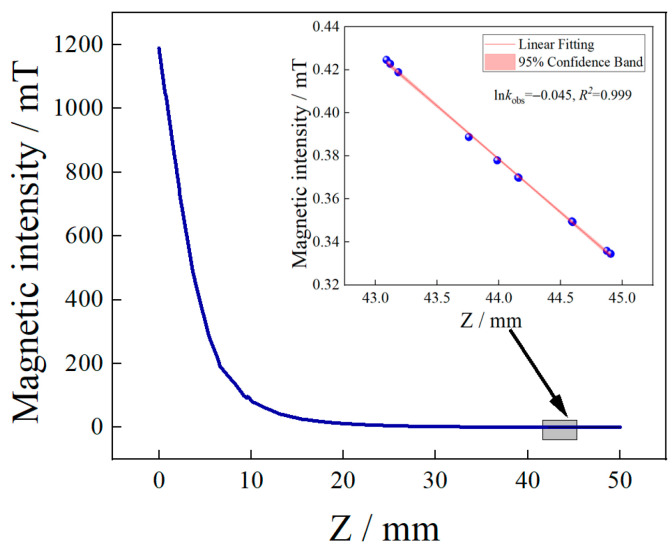
The strength of the magnetic field emanating from the center of a cylindrical magnet.

**Figure 3 micromachines-14-00938-f003:**
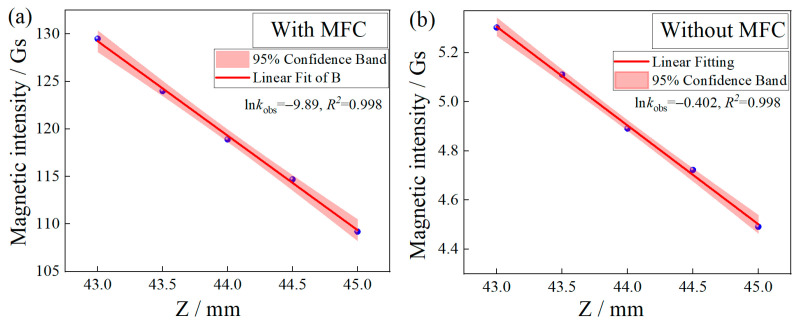
(**a**) The relationship between magnetic field intensity and position change in the presence of MFC. (**b**) Relationship between magnetic field intensity and position change in the absence of MFC.

**Figure 4 micromachines-14-00938-f004:**
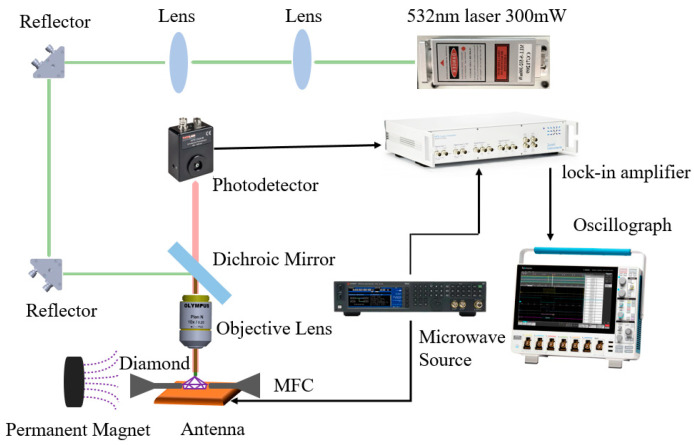
Schematic diagram of the experimental test system.

**Figure 5 micromachines-14-00938-f005:**
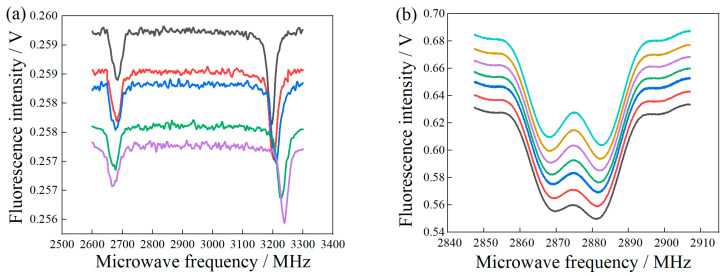
(**a**) Variation of ODMR resonant frequency in the presence of MFC. (**b**) Variation of ODMR resonant frequency in the absence of MFC. (**c**) Relationship between frequency variation and displacement in structures with and without MFC. (**d**) The relationship between frequency change and displacement in a structure without MFC.

**Figure 6 micromachines-14-00938-f006:**
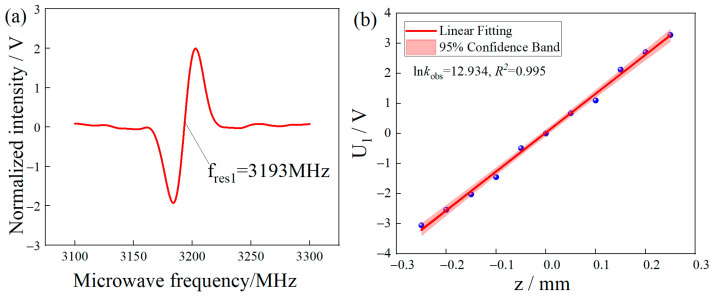
(**a**) Output curve of lock-in amplifier with MFC. (**b**) The relationship curve between displacement and voltage of MFC. (**c**) Output curve of phase-locked amplifier without MFC. (**d**) Relationship curve between displacement and voltage without MFC.

**Figure 7 micromachines-14-00938-f007:**
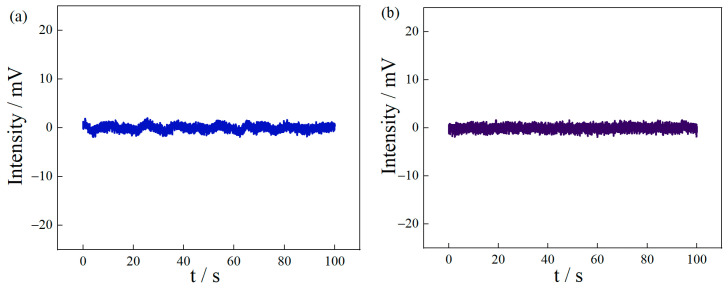
(**a**) MFC system output noise waveform under fixed frequency signal. (**b**) In the fixed frequency signal input, there is no MFC output noise waveform.

## Data Availability

Not applicable.

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
