# Peer review of "Research on Micro-Displacement Measurement Accuracy Enhancement Method Based on Ensemble NV Color Center"

_micromachines, 2023, doi:10.3390/mi14050938_

Round 1
Reviewer 1 Report
NV center magnetometer have a wide range of applications in engineering, science, and medicine. Their sensitivity depends on the quality of the diamond material, on the parameters of the excitation light, the efficiency of extracting the fluorescence light that is emitted by the NV centers and other parameters. In particular, further effort is necessary to provide a homogeneous offset magnetic field in addition to a sufficiently strong and homogeneous microwave field.
The authors present a micro-displacement test system by combining the correlation between magnetic flux concentrator, permanent magnet and micro-displacement. As an analogy, they propose to compare their system with that proposed in [Wang C, et al. Opt Lasers Eng. 86 (2016)125] with a sensitivity of 8.8 mV/µm. The authors of the manuscript claim that the system they proposed makes it possible to achieve a sensitivity of 12.934 V/mm by using a magnetic flux concentrator and increasing the resolution of the displacement test system about 26 times to 25 nm. It should be noted that the use of a magnetic flux concentrator is a standard technique in the manufacture of an NV center magnetometer. In particular, the use of ferrite flux concentrator in [Fescenko I. et al. Phys. Rev. Research 2 (2020) 023394] made it possible to realize a ∼250-fold increase of the magnetic field amplitude within the diamond. Note that the use of a magnetic concentrator is equivalent to amplifying the magnetic field, while not actually improving the magnetic sensitivity of the NV center magnetometer.
Unfortunately, in the manuscript there is practically no comparison of the results achieved with the current state in the development and creation of NV center magnetometers and magnetic flux concentrators. I consider it necessary to supplement the text of the manuscript with such a comparison. It is necessary to describe in detail the fundamental difference between the proposed micro-displacement test system and existing analogues. Please, write what exactly the advantages of your method are - improved performance, low cost, simplicity, stability, etc.
Usually the sensitivity of the NV center magnetometers is measured in nT/Hz1/2. For example, at Jinan University the magnetic field detection sensitivity of the probe is significantly enhanced to 0.57 nT/ Hz1/2@ 1Hz [Chen Y. et al. ACS sensors 7(2022) 3660]. If you're having trouble with this, please provide data on sensitivity measured in mV/µm for several papers besides [Wang C, et al. Opt Lasers Eng. 86 (2016)125].
To approximate the dependencies, the authors use a linear function, although in some cases (for example, in Figure 3) this is not entirely correct. It would be better to plot the accuracy of the measurements directly on the graphs, showing the confidence interval. It is necessary to bring the declared parameters in line with the measurement accuracy. In particular, judging by the figures in the article, the value of sensitivity is apparently equal to 13 or 12.9 V/mm. The same applies to the effect of using a magnetic flux concentrator. The text says that the change occurs by a factor of 26, although judging by the first three measurement points in Figure 4, this value does not exceed 25.
Author Response
Thank you for your recognition of our work. Your careful and professional comments helped us correct the errors in the manuscript. The corresponding revises to this manuscript based on your comments were made in our revision.
Please see the attachment for the reply to the question.
Reviewer 2 Report
The Introduction paragraph that reviews the state-of-the-art and recent achievements related to the discussed topic looks a little-bit short, but mainly covers the essence of the current level reached in this research are.
Usually, in a journal paper a paragraph title never is followed directly by a figure caption (see paragraph 2.) It is recommended the introduction of the figure with 1-2 sentences, than may be inserted the figure.
The used mathematical background is adequate with accurate edited equations.
In Fig. 5 it is given the schematic block diagram of the experimental test system. To insert a picture of the used experimental system (by indicating its component modules) looks also welcome.
The references list is up to dated and adequate.
The English spell checking of the whole paper is also recommended, there are minor grammar errors.
As a general conclusion, the paper describes a meritorious research effort with concrete results and contribution.
Author Response

(The authors gave the same response as above.)

Round 2
Reviewer 1 Report
The authors of the manuscript made some changes to the text and figures. However, they partially ignored one significant remark - “…., judging by the figures in the article, the value of sensitivity is apparently equal to 13 or 12.9 V/mm”. The same applies to the effect of using a magnetic flux concentrator. The text says that the change occurs by a factor of 26, although judging by the first three measurement points in Figure 4, this value does not exceed 25. ” It's a little sad to have to explain in a review of a good article the basics of experimental data processing. The fact is that if you define average speed as the ratio of the distance traveled to time and measure distance with an accuracy of 0.2%, and time with an accuracy of 0.1%, then the speed is calculated as their ratio with an accuracy of 0.3%.
In your system, in addition to the scatter in measurements, which you approximate with a linear function with a confidence of 95%, there is a spatially inhomogeneous dependence of “The strength of the magnetic field emanating from the center of a cylindrical magnet” (Figure 2), which you also approximate with a linear function. I don't know the motives for which you insist on keeping values like “sensitivity of the displacement data is 12.934 V/mm”, “resolution of the displacement test system without MFC is 659 nm” and others like this in the text. I believe that the publication of calculated data without taking into account the experimental errors in their determination reduces confidence not only in the results obtained, but will also not correspond to the status of an authoritative scientific journal.
Author Response
We would like to acknowledge the editor and reviewer for having spent time on handling and reviewing this manuscript. We studied carefully your insightful comments, and a number of necessary changes were made and highlighted in red in the manuscript based on your comments. We believe that these changes can fully address all concerns of reviews.
